# Research on Behavioral Decision-Making of Subjects on Cultivated Land Conservation under the Goal of Carbon Neutrality

Yun Teng [1,2] and Peiwen Lin [3,*]

1    College of Engineering, Northeast Agricultural University, Harbin 150030, China
2    Agricultural and Forestry Economics and Management Postdoctoral Station, Northeast Agricultural University, Harbin 150030, China
3    School of Business, Xinjiang University, Urumqi 830046, China
*    Correspondence: linpeiwen2021@163.com

**Abstract:** Protecting cultivated land is an urgent mitigation measure for China to reconcile the contradiction between food safety and carbon neutrality. In the context of carbon neutrality, this paper constructs an evolutionary game model among local governments, agricultural technology service organizations (ATSOs), and farmers based on China's cultivated black land, and discusses the factors influencing the strategy choice of each stakeholder group and the final form of evolutionary stabilization strategies adopted by each stakeholder from the perspective of agricultural extension. Through numerical simulations, we reveal that two stable situations exist in the current state of protection of cultivated black land in China: full subject participation and government subject participation only. In order to achieve the optimal realization of the dynamic equilibrium of the three parties, we identify the key issues of cultivated land protection (CLP) and put forward reasonable suggestions, which are summarized as follows: (1) prohibit the excessive subsidies to farmers, and keep the appropriate subsidies at 100~140 CNY/mu to help the protection of cultivated land, if more than 140 CNY/mu is not conducive to the participation of local governments in CLP (mu, a Chinese unit of land measurement that equals to 1/15 a hectare); (2) an increase in the farmers' fines has a dampening effect on farmers digging black soil, and the game model achieves the ideal equilibrium when it reaches 10 billion CNY, which can be implemented as a long-term cultivated land protection policy; (3) maintaining the incentive fund at 5 billion CNY provides the greatest incentive for ATSOs to promote low-carbon agricultural technologies (LCAT), while the production trusteeship subsidies has no beneficial impact on ATSOs; (4) reducing production trusteeship costs and not increasing service charges is the most effective way of incentivizing ATSOs to promote LCAT. This means the service fee is maintained at 400 CNY/mu and the service cost is reduced to 308 CNY/mu. This study reveals the inner mechanism of CLP, provides a theoretical basis for the promotion of CLP technology, and proposes effective cultivated land protection suggestions, aiming to improve the overall implementation effect of CLP in China and help carbon neutrality.

**Keywords:** cultivated land protection; carbon neutrality; evolutionary game; agricultural technology service organization; black land

## 1. Introduction

It is imperative for China to promote the process of carbon neutrality. Excessive carbon emissions have led to drastic changes in the global climate [1], posing severe challenges to human development, including extreme climates, ecological fragility, and reduced food production, even threatening human survival [2,3]. At the 21st United Nations Climate Change Conference held in Paris, 196 sovereign countries adopted the Paris Agreement and reached a consensus on achieving net-zero global greenhouse gas emissions, marking the reduction in greenhouse gas emissions as a global goal. Agriculture is the second largest

source of carbon emissions [4,5], accounting for 10–12% of global carbon emissions [6,7]. In particular, China's agricultural greenhouse gas emissions account for 17% of the country's carbon emissions [8], which has far exceeded the world average and is still growing. As a large agricultural and populous country, China needs to rapidly develop its agriculture to meet the growing demand for food. However, China's total carbon emissions are still far from peaking, and more agricultural production has led to a significant increase in greenhouse gas emissions [9]. Under these conditions, China has committed to reaching a carbon peak by 2030 and achieving carbon neutrality by 2060. It reflects not only China's responsibilities but also the inherent need to achieve green transformation and sustainability. Meanwhile, China will face enormous pressure to achieve carbon neutrality [10].

Low-carbon technologies give agriculture the unique ability to achieve carbon neutrality. Different from other industries, agriculture has the dual attributes of a carbon source and carbon sink, as well as the advantages and potential to achieve carbon neutrality within the industry. Low-carbon technologies, such as controlling agricultural inputs [11,12] and reducing wasteland reclamation [13,14], can help agroecosystems transform from carbon sources to carbon sinks [15,16], enabling agroecosystems to offset 80% of the greenhouse gas emissions from agriculture [17]. Therefore, low-carbon technologies are considered to be an effective way to reduce agricultural carbon emissions [18]. However, there is little discussion of LCAT [19]. The extensive implementation of agricultural intensification in China in the past few decades [20] has led to the extensive use of chemical fertilizers, pesticides, and agricultural film, which has increased agricultural carbon emissions. Compared to the progress in developed countries, LCAT has not been widely adopted in China. China urgently needs to develop LCAT, including improving the technical level of farmers' cultivated land [21], reducing the use of chemical fertilizers and pesticides, and cooperating with conservation tillage techniques such as deep plowing and less tillage [22]. In this way, China can maximize the reduction in cultivated land carbon emissions, and strengthen the carbon sequestration capacity of cultivated land. The application of conservation tillage technologies to cultivated land has both ecological and economic benefits and is an effective means to coordinate the contradiction between food security and carbon neutrality.

CLP is the primary goal to ensure food security and achieve carbon neutrality. As an important ecological module of agriculture, cultivated land is both the main carbon source and carbon sequestration unit, and makes a fundamental contribution to achieving carbon neutrality. Countries have adopted a series of measures to implement CLP, such as the agricultural revitalization regional system in South Korea, which strictly restricts land diversion, and Japan, which also uses land-use control to protect cultivated land. Developed countries focus on increasing economic compensation and incentives to protect the quality of cultivated land, such as the UK's environmentally friendly farming policy, which aims to promote CLP by subsidizing farmers. Although China has implemented the most stringent basic policy of CLP in the world [23], the effect of CLP is not optimistic [24]. Long-term over-fertilization [25], loss of soil organic content (SOC) [26], and conversion of high-quality cultivated land into construction land [27] have resulted in a continuous decline in the quality of cultivated land. As the "ballast stone" for food and the most important source of agricultural carbon emissions in China, the large area of black soil in Northeast China, with a high organic matter content, is an important part of agricultural carbon neutrality and should be given special attention. The decline in SOC not only leads to the degradation of the quality of arable black land [28] but also threatens food security and releases a large amount of carbon. China has released the *National Blackland Protection Project Implementation Plan (2021–2025)* to strengthen the promotion and application of protection tillage technologies and call for the active participation of farmers and other entities in farmland protection.

Protection of cultivated black land has become a major effort for China to achieve carbon neutrality. The most important aspects are CLP technology and cultivated land quality [29]. However, the failure of CLP in China in the past stemmed from the conflicting interests of different participants in the process of farmland protection [30]. The participants

were more concerned with maximizing their interests than farmland protection [31]. To promote conservation tillage technology, China needs to call on farmers and other actors to actively participate in CLP. It is necessary and important to find the main players in CLP and to coordinate the conflicting interests among subjects. CLP is a complex behavioral game process, involving farmers, ATSOs, local governments, and other stakeholders with different interests, expressions, and conflicts of interest, and it is necessary and important to coordinate the conflicts of interest among subjects. Therefore, this paper adopts the evolutionary game approach to simulate and analyze the CLP in Chinese black soil as an example.

The purpose of this paper is to coordinate the interests of each subject, analyze the behavioral decisions of each subject, and identify the key issues of interest game focus and CLP. By exploring the interaction mechanisms and evolutionary trends of the strategic choices of each subject under different parameters, we will evaluate government policies and consider how to improve them, so as to provide theoretical guidance for CLP and continuously optimize China's CLP strategy, which will serve as a reference for other countries and regions in the world. This paper also focuses on the contribution of CLP in achieving carbon neutrality and the impact of carbon neutrality on the implementation of CLP, with a view to improving the quality of cultivated land, reducing agricultural carbon emissions, and achieving carbon neutrality as soon as possible.

## 2. Literature Review

Numerous scholars have made rich research results in CLP. In the quantity of cultivated land, scholars have delineated the area of basic farmland to inform farmland protection legislation [32,33]. The agricultural land reserve in British Columbia is considered to be the earliest international example of relatively successful monitoring legislation [34]. With accelerated urbanization, CLP in suburban areas has also received attention from some scholars, such as the changes brought by the French farmland protection program in managing the loss of cultivated land in suburban areas [35] and the negative impact of cultivated land diversion on farmers' assets in suburban areas in Ethiopia [36]. However, most of the previous studies on farmland protection tend to ensure the amount of farmland to prevent its loss, abandonment [37], and diversion, and are not applicable to studying the use of low-yielding and non-diversion farmland by local governments. The lack of central government regulation makes local governments and farmers unaware of the importance of conserving cropland and also tends to lead to non-cooperation between local government officials and farmers [38]. Conflicting interests exist between farmers, local governments, and the central government in CLP [39]. Therefore, more scholars have focused on the stakeholders of CLP and made pioneering studies. Barham et al. argued that farmers are key players in CLP [40] and have a negative impact on CLP [41]; Skinner et al. argued that the government's attitude has a direct impact on CLP [42]. Since the social acceptability of the participating stakeholders varies [43], the stability of CLP can only be achieved by coordinating the interests of all parties. Many studies have constructed theoretical or empirical analysis models based on game theory to analyze the conflict of interests among stakeholders. For example, Zellner et al. used the prisoner's dilemma theory to analyze the conflict of interests in fallow cropland management and proposed to achieve desired land management through subsidies [44]. However, most of these studies are based on static games and the assumption of perfect rationality of participants, which is not consistent with reality. Therefore, we need to apply evolutionary games to model the strategy changes of participants in reality.

Evolutionary game theory (EGT) combines game theoretical analysis with the analysis of dynamic evolutionary processes, assuming that human rationality is limited [45] and that the complete information condition is unnecessary, which is more realistic than traditional games. EGT uses trial-and-error methods to seek game equilibrium, and the strategies of both parties eventually converge to evolutionary stable strategies (ESS) [46], which is in line with the laws of evolution, and therefore was initially widely used in the field

of biology [47,48]. To analyze the strategic interactions of multiple parties, EGT was introduced for decades [49] and has been applied to many other fields. For example, in the field of carbon trading, EGT is used to explore the impact of carbon emissions on the cooperative behavior of solar power plants and coal-fired thermal power plants from the perspective of energy producers [50]; in the field of supply chains, EGT is used to explore the behavior of low-carbon supply chain firms and strategic issues related to government low-carbon policies and emerging low-carbon markets [51]; in the field of environmental regulation, [52] constructed a tripartite evolutionary game model of regulators, energy companies and whistleblowers to study the energy regulatory system in China; in the field of economics, EGT methods are often used to predict future development trends [53]; and in the field of energy, EGT is used to analyze the behavioral decisions among stakeholders in promoting clean energy technologies [54]. In addition, EGT can be applied to environmental impact assessment cases to investigate the rationality of the decisions that occur [55]. Other disciplines have also used EGT methods to address forecasting or management problems [56,57].

EGT has been applied to research on cropland protection and management involving multiple stakeholders, such as fallow land protection [58], cropland abandonment management [59], cropland allocation [60], spatial land-use planning [61], and land hoarding [62]. All of the above studies are groundbreaking and show that EGT can effectively elucidate the contradictory focus and decision evolutionary features of various stakeholders in CLP. Due to information asymmetry and different objectives and interests [63], the protection stakeholders may exhibit limited rationality through multi-stage games, which directly or indirectly affect the protection situation of farmland and thus the carbon-neutrality process. The application of EGT helps to understand the dynamic process of subject evolution and explain how to reach a steady state, so as to provide more reasonable policy recommendations.

Although CLP is a key concern, no studies have been conducted on EGT from the perspective of agricultural extension. Existing studies focus more on the behavioral decisions of subjects such as farmers or enterprises but tend to ignore service subjects such as cooperatives, agricultural extension agencies, and private farm advisors [64]. As individuals implementing the promotion of arable land protection [40], it is difficult for farmers to abandon the proven conventional agricultural business model [65] and choose LCAT before they see the benefits. It has been suggested that agricultural technology promotion is an important tool to promote the development of agricultural modernization [66]. Agrotechnical service subjects, on the other hand, are the main subjects of promoting modern agricultural technologies and are responsible for organizing farmers' participation in extension training and trusteeship production, which releases the potential of farmers [67] and enables sustainable production of arable land and reduces carbon emissions. Therefore, it is necessary to conduct specialized research on the main body of ATSOs. In addition, as countries pay more attention to carbon neutrality, achieving carbon neutrality has become an indispensable macro power for the implementation of CLP, but there is little literature exploring the contribution of CLP in achieving carbon neutrality, and even less research considering the impact of carbon-neutral factors on CLP in EGT studies.

In summary, this paper adds ATSOs to the evolutionary game model, fills the research gap of the behavioral decision of ATSOs in CLP, focuses on revealing the intrinsic mechanism of stakeholders in promoting agricultural technology, and conducts simulations. By elaborating on the correlation between CLP and carbon neutrality, the influence of carbon-neutral factors on CLP is fully considered, thus enriching the study of the behavioral decision making of local government and farmers' subjects in CLP under the carbon-neutrality objective.

## 3. Game Model

### 3.1. Main Body Description

Under the goal of carbon neutrality, the realization of CLP is the result of continuous interaction, repeated friction, and complex negotiations among local governments, ATSOs, and farmers. Therefore, this paper needs to first discuss the behavioral decision-making characteristics of the abovementioned subjects and then explain the relationship among the stakeholders. The relationship between the three subjects is shown in Figure 1.

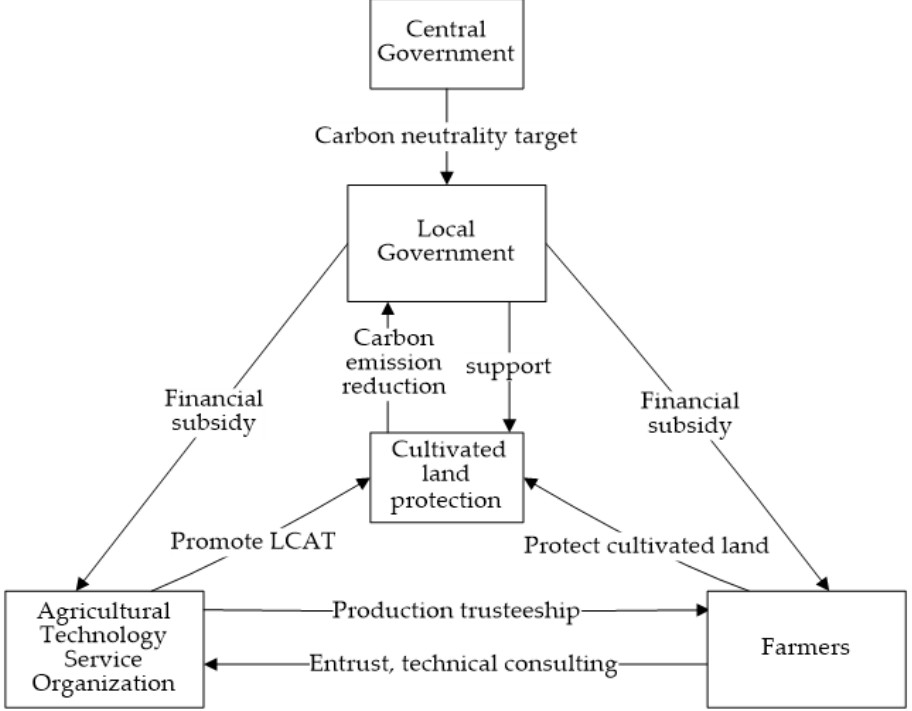

**Figure 1.** Stakeholders' relationship diagram.

Local governments are supervised by the central government and are the actual executors of farmland protection policies. To achieve carbon-neutrality targets, local governments have established agricultural carbon compensation mechanisms through incentive subsidies and carbon trading, supported ATSOs to promote LCAT [68], and guided farmers to actively participate in farmland protection. However, the current situation of "large grain-producing counties and financially poor counties" has not been fundamentally resolved. Local governments not only face the assessment of carbon-neutrality indicators but also face the pressure of economic development. At the same time, they need to coordinate the contradiction between grain production and economic construction, which makes it difficult for most local government policies to achieve the expected effect of the central government. Therefore, to avoid the loss of short-term economic growth due to the protection of cultivated land [69], local governments may be less concerned about the protection of cultivated land but are more inclined to pursue economic development, and even occupy agricultural cultivated land for urban construction. The opportunity cost is passed on to other regions. The decision-making behavior of local governments can be classified as supporting or not supporting CLP.

ATSOs are the core power to promote CLP. According to service purpose, profitability and organizational structure, it can be classified into dichotomy [70], three points [71], and five points [72]. At present, China has formed an agricultural technology extension pattern of "one major and multiple", with government-led public welfare agricultural technology extension institutions as the leading ones, supplemented by various social service organizations (including agricultural research organizations, private organizations, and collective economic organizations). The public welfare and commercial agricultural

technology extension services are integrated and should be considered as the same subject to analyze. Although public agencies are the main force in promoting LCAT, their single services, insufficient government subsidies, and backward technology lead to the phenomenon of promoting high-carbon conventional agricultural technology (HCAT) to farmers in terms of in-production services. With the rise in socialized agricultural technology service groups, the agricultural trusteeship production mode, which is mainly based on whole-process trusteeship and supplemented by semi-trusteeship, effectively connects small farmers, realizes large-scale agricultural operations, and drives the promotion and application of LCAT. However, the development of socialized agricultural technology service groups is still insufficient. To obtain more farmer groups, some organizations apply HCAT to manage production to reduce costs but increase carbon emissions and hinder the realization of carbon neutrality. Therefore, the decision-making of ATSOs on the promotion of agricultural technology can be divided into the promotion of LCAT or HCAT.

Farmers are the direct implementers of CLP, and they are also an important subject to enhance the carbon-neutrality effect. The higher the farmers' expectation of the price of low-carbon agricultural products, the higher the willingness to adopt low-carbon production [73]. At this time, farmers will actively seek the services and technologies of ATSOs to help protect cultivated land. However, it is unrealistic for farmers to rely on the conscientiousness of protecting cultivated land to complete the transformation of low-carbon agriculture under the circumstances of the government's laissez-faire. Farmers who lack awareness of protection will dig and destroy black land, which will reduce the carbon sequestration capacity of cultivated land and aggravate carbon emissions. Affected by short-sighted interests, farmers prefer conventional farming to low-carbon farming with reduced profits in the short term. However, HCAT, such as excessive application of chemical fertilizers and a large number of pesticides, will lead to a decline in the quality of cultivated land, which can be considered a predatory production technique that overdraws the future fertility of the soil. The adoption of agricultural technology by farmers is a dynamic behavior influenced by internal and external factors [74]. The effective publicity and improvement of subsidy policies by local governments will help to increase farmers' enthusiasm for farmland protection. The active services of ATSOs can assist farmers to participate in the protection of cultivated land, promote the popularization of LCAT, and play the role of cultivated land in enhancing the carbon-neutrality effect. Therefore, farmers' decisions in farmland protection can be classified as protecting or not protecting farmland.

*3.2. Model Hypothesis*

**Hypothesis 1.** *Each stakeholder perceives and selects strategies based on bounded rationality and follows the principle of utility maximization during the game.*

**Hypothesis 2.** *Local governments refer to grassroots governments at the township, town, and county levels. Considering carbon-neutrality targets, political achievements, environmental governance, farmland subsidies, and other factors, local governments incentivize ATSOs and farmers in the form of financial subsidies, carbon taxes, and fines to encourage the promotion of LCAT to achieve carbon-neutrality goals. The local government's decision is divided into "support" or "not support".*

**Hypothesis 3.** *ATSOs provide technology and services for farmers, including public welfare agricultural technology extension agencies and socialized service organizations. It is affected by various factors such as environmental benefits brought by carbon neutrality, government policies, organizational development benefits, and credibility benefits. The decision-making choice is to promote "HCAT" or "LCAT".*

**Hypothesis 4.** *Farmers are bounded rational individuals, including small and medium farms, agricultural individuals, and family farmers, and are "rational economic people". Affected by factors such as agricultural technology level, awareness of CLP, government subsidies, and personal interests, farmers consider entrusting ATSOs to assist in the transformation from conventional agriculture to low-carbon*

*agriculture, responsibility for the profits and losses of cultivated land production, and decide whether to "protect" or "not protect"* [74].

The proportion of local governments "support" and "not support" is x and 1 − x, respectively. The proportion of ATSOs choosing to promote "LCAT" or "HCAT" is y and 1 − y, respectively. The proportion of farmers choosing "protect" and "not protect" cultivated land is z and 1 − z, respectively, and x, y, z ∈ [0, 1].

*3.3. Parameter Assumptions*

3.3.1. The Local Governments' Related Benefit Assumptions

When local governments choose "support", the subsidies for the reform and construction of agricultural technology extension system (including the subsidy for the capacity building of agricultural technical personnel, for agricultural technology extension service and project implementation, and for agricultural technology demonstration) are $C_1$; the incentive dedicated funds set up by local governments for ATSOs promoting LCAT are $C_2$; and the subsidies for production trusteeship of the ATSOs are $C_3$. The subsidies given to farmers who protect cultivated land are $B_1$ (including farmland productivity, deep loosening soil preparation, organic fertilizer, straw counters-field and other subsidies); and you charge a fine of $B_2$ to farmers who do not protect their cultivated land (the behavior refers to the destruction and digging of black soil, blind reclamation and predatory production etc., resulting in carbon emission). The cost of governance and publicity for the deterioration of cultivated land environment caused by farmers' failure to protect cultivated land is $R$. The performance benefits and social benefits brought by the increase of grain yield are $H_1$; and the long-term comprehensive benefits gained by governments from protecting farmland or governance to meet carbon emission targets and enhance carbon neutrality are $H_2$ (including ecological benefits brought by carbon-neutrality effect, future potential and benefits brought by the improvement of cultivated land quality, economic benefits from a booming carbon markets).

When local governments choose "not support", they must also support the reform and construction of the agricultural technology extension system with subsidy $C_1$, but do not set up incentive-dedicated funds and do not issue special subsidies for production trusteeship. The economic construction benefits obtained from transforming agricultural land into economic land are $H_3$. The long-term benefit loss caused by farmers' failure to protect cultivated land and local governments' inaction is $V$ (including the long-term decline of carbon-neutrality effects, such as grain reduction, increase of environmental governance cost and decrease of government credibility).

3.3.2. The ATSOs' Related Benefit Assumptions

When ATSOs choose to promote "LCAT", the working and operating expenses are $C_4$, subsidies for reform and construction are $C_1$, incentive dedicated funds are $C_2$, and the production trusteeship subsidies are $C_3$; the capacity building costs for agricultural technicians are $T$ (such as talent incentive, the regular organization for agricultural technicians to learn knowledge of LCAT, purchase of modern advanced instruments and equipment). The expenditures related to the demonstration and promotion of LCAT in villages are $M_1$ (including pre-production purchase of agricultural materials and technical services, high-quality technical guidance such as deep plowing and deep soaping in production, and post-production grain collection and storage services). The costs of ATSOs for providing farmers low carbon production hosting services are $M_2$ (including monitoring of crop seedlings and quality of farmland, monitoring of agro-ecological environment and use of agricultural input, etc.). The low carbon production trusteeship fee charged to farmers is $F_1$, and the long-term development benefit obtained due to the carbon-neutrality effect is $D_1$ (including government support and social recognition, increase in market share and competitiveness, etc.).

When ATSOs choose to popularize "HCAT", the working and operating expenses are $C_4$, subsidies for reform and construction are $C_1$, and the production trusteeship subsidies

are $C_3$; The related expenditure for entering villages to start conventional agricultural technology popularization is $M_3$ (the expenditure on yield-enhancing technologies in the mid-production stage). To gain competitiveness and improve profits, the cost for commercial ATSO to adopt HCAT to offer production trusteeship to farmers is $M_4$, and the trusteeship service fee under HCAT charged to farmers is $F_2$. The adverse impact of the promotion of HCAT on long-term development is $N$ (failure to comply with the goal of carbon neutrality, resulting in loss of reputation; failure to update knowledge levels, resulting in a decrease in the number of farmers' entrustment; a decline in carbon trading income and a decline in competitiveness, etc.).

3.3.3. The Farmers' Related Benefit Assumptions

When farmers choose "protect", the subsidy from the local government is $B_1$. If LCAT is lacking, the protective production cost paid by farmers for self-planting is $A_2$ (including the cost of technical consultation, labor, low-carbon agricultural materials, and equipment). The crop income earned by farmers for self-planting is $K_3$. Farmers only need to pay $F_1$ for trusteeship production when ATSOs promote LCAT. Because the trusteeship production adopts LCAT to improve the organic matter content of black soil, the potential for future crop yield increase is $L_1$ (such as obtaining high-quality agricultural products to increase income, and the production of low-carbon products is favored by the market). The enhanced carbon sink capacity of the cultivated land system will bring the carbon-neutrality benefit to farmers as $L_2$ (including direct and indirect benefits brought by carbon neutralization in the long term).

When farmers choose "not protect", the production cost of self-planting under HCAT is $A_1$ (including input of conventional agricultural materials, labor and farming machinery, etc.). If the ATSOs adopt HCAT to provide production trusteeship, the service charges are $F_2$. The yield-enhancing technology of HCAT brings farmers a crop benefit of $K_1$, but they cannot receive government subsidies for protecting cultivated land. Due to the illegal digging of black soil, the farmers obtained $K_2$ illegal profits but faced a $B_2$ fine from the local government. The loss of farmland fertility caused by long-term predatory production is $Q_1$ (including a decline in agricultural product quality and yield, etc.); the adverse impact of the increase in carbon emissions caused by conventional agricultural production is $Q_2$.

Based on the above assumptions, the return matrix of the three-party game constructed by the return portfolio is shown in Table 1.

**Table 1.** The tripartite game income matrix.

| Strategy | Benefits of Local Governments | Benefits of ATSOs | Benefits of Farmers |
|---|---|---|---|
| (0,0,0) | $H_3 - C_1 - V$ | $C_1 - C_4 - M_3 - N + F_2 - M_4$ | $K_1 + K_2 - Q_1 - Q_2 - F_2$ |
| (0,0,1) | $H_3 - C_1$ | $C_1 - C_4 - M_3 - N$ | $L_1 + L_2 - A_2 + K_3$ |
| (0,1,0) | $H_3 - C_1 - V$ | $C_1 - C_4 - T - M_1 + D_1$ | $K_1 + K_2 - Q_1 - Q_2 - A_1$ |
| (0,1,1) | $H_3 - C_1$ | $C_1 - C_4 - T - M_1 - M_2 + F_1 + D_1$ | $L_1 + L_2 - F_1 + K_3$ |
| (1,0,0) | $H_1 + H_2 - C_1 - R + B_2 - C_3$ | $C_1 + C_3 - C_4 - M_3 - N + F_2 - M_4$ | $K_1 + K_2 - Q_1 - Q_2 - F_2 - B_2$ |
| (1,0,1) | $H_1 + H_2 - C_1 - B_1$ | $C_1 - C_4 - M_3 - N$ | $B_1 - A_2 + K_3 + L_1 + L_2$ |
| (1,1,0) | $H_1 + H_2 - C_1 - R + B_2 - C_2$ | $C_1 + C_2 - C_4 - T - M_1 + D_1$ | $K_1 + K_2 - Q_1 - Q_2 - A_1 - B_2$ |
| (1,1,1) | $H_1 + H_2 - C_1 - C_3 - C_2 - B_1$ | $C_1 + C_2 + C_3 - C_4 - T - M_1 - M_2 + F_1 + D_1$ | $B_1 - F_1 + K_3 + L_1 + L_2$ |

## 4. Equilibrium and Stability Analysis of Evolutionary Game Models

The proportions of the various subsequent pure strategies used by all individuals are used to represent the mixed strategies in the game model. According to Table 1, this paper carries out the strategy equilibrium analysis of the evolutionary game model.

### 4.1. Equilibrium Analysis

4.1.1. Equilibrium Analysis of Local Governments

Local governments' expected income by adopting the "support" and "not support" strategies are $V_x$ and $V_{1-x}$.

$$V_x = B_2 - C_1 - C_3 + H_1 + H_2 - R - z(B_1 + B_2 - C_3 - R) + y(C_3 - C_2) - 2C_3yz \quad (1)$$

$$V_{1-x} = H_3 - C_1 - V + zV \quad (2)$$

The local governments' decision-making replication dynamic equation is as follows:

$$\begin{aligned} F(x) &= \frac{dx}{dt} = x(1-x)(V_x - V_{1-x}) \\ &= x(x-1)(C_3 - B_2 + R - H_1 - H_2 + H_3 - V + z(B_1 + B_2 - R + V - C_3) + y(C_2 - C_3) + 2C_3yz) \end{aligned} \quad (3)$$

The first derivative for $F(x)$ of x is as follows:

$$\frac{d(F(x))}{dx} = (2x-1)(C_3 - B_2 + R - H_1 - H_2 + H_3 - V + z(B_1 + B_2 - R + V - C_3) + y(C_2 - C_3) + 2C_3yz) \quad (4)$$

The stable state of the principal decision must satisfy $F(x) = 0$ and $d(F(x))/dx < 0$ [75].

When $y = \dfrac{B_2 - C_3 - R + H_1 + H_2 - H_3 + V + z(-B_1 - B_2 + C_3 + R - V)}{C_2 - C_3 + 2C_3z} = y^*$,

$F(x) = 0$ and $d(F(x))/dx = 0$, which means that the local government cannot determine the stability strategy.

When $y \neq y^*$, the evolutionarily stable point may be $x = 0$ or $x = 1$. When $y > y^*$, $\frac{dF(x)}{dx}\Big|_{x=0} < 0$, $x = 0$ is the Evolutionarily Stable Strategy (ESS) of local governments. Conversely, $x = 1$ is ESS. The phase diagram of local governments' evolution is shown in Figure 2a.

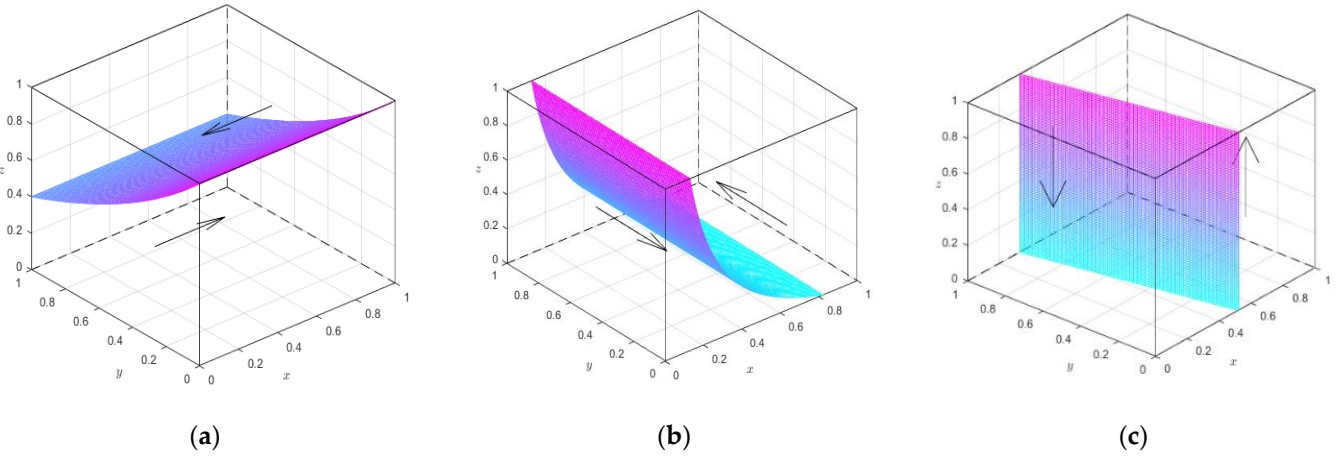

(a)                                                           (b)                                                           (c)

**Figure 2.** The phase diagram of the three stakeholders' dynamic evolution: (**a**) description of the local governments' dynamic evolution; (**b**) description of the local ATSOs' dynamic evolution; (**c**) description of the farmers' dynamic evolution. The arrow indicates the evolutionary direction of the initial stakeholders falling into the region where the arrow is located.

4.1.2. Equilibrium Analysis of ATSOs

ATSOs' expected income by adopting the "LCAT" and "HCAT" strategies are $W_y$ and $W_{1-y}$.

$$W_y = C_1 - C_4 + D_1 - M_1 - T + C_2x + z(F_1 - M_2) + C_3xz \quad (5)$$

$$W_{1-y} = C_1 - C_4 + F_2 - M_3 - M_4 - N + C_3x + z(M_4 - F_2) - C_3xz \quad (6)$$

The ATSOs' decision-making replication dynamic equation is as follows:

$$\begin{aligned} F(y) &= \frac{dy}{dt} = y(1-y)(W_y - W_{1-y}) \\ &= y(1-y)(D_1 - F_2 - M_1 + M_3 + M_4 - T + N + z(F_1 + F_2 - M_2 - M_4) + x(C_2 - C_3) + 2C_3xz) \end{aligned} \quad (7)$$

The first derivative for $F(y)$ of $y$ is as follow:

$$\frac{d(F(y))}{dy} = (1-2y)(D_1 - F_2 - M_1 + M_3 + M_4 - T + N + z(F_1 + F_2 - M_2 - M_4) + x(C_2 - C_3) + 2C_3xz) \quad (8)$$

When $x = \frac{-(D_1 - F_2 - M_1 + M_3 + M_4 + N - T + z(F_1 + F_2 - M_2 - M_4))}{C_2 - C_3 + 2C_3z} = x^*$, $F(y) = 0$
and $d(F(y))/dy = 0$, which means the ATSOs cannot determine the stability strategy.

When $x \neq x^*$, the evolutionarily stable point may be $y = 0$ or $y = 1$. When $x > x^*$, $\frac{dF(y)}{dy}\Big|_{y=1} < 0$, $y = 1$ is the ESS of ATSOs. Conversely, $y = 0$ is ESS. The phase diagram of ATSOs' evolution is shown in Figure 2b.

### 4.1.3. Equilibrium Analysis of Farmers

Farmers' expected income by adopting the "protect" and "not protect" strategies are $U_z$ and $U_{1-z}$.

$$U_z = K_3 - A_2 + L_1 + L_2 + B_1x + y(A_2 - F_1) \quad (9)$$

$$U_{1-z} = K_1 - F_2 + K_2 - Q_1 - Q_2 - B_2x + y(F_2 - A_1) \quad (10)$$

The farmers' decision-making replication dynamic equation is as follows:

$$F(z) = \frac{dz}{dt} = z(1-z)(U_z - U_{1-z})$$
$$= z(1-z)(F_2 - A_2 - K_1 - K_2 + K_3 + L_1 + L_2 + Q_1 + Q_2 + y(A_1 + A_2 - F_1 - F_2) + x(B_1 + B_2)) \quad (11)$$

The first derivative for $F(z)$ of $y$ is as follows:

$$\frac{d(F(z))}{dz} = (1-2z)(F_2 - A_2 - K_1 - K_2 + K_3 + L_1 + L_2 + Q_1 + Q_2 + y(A_1 + A_2 - F_1 - F_2) + x(B_1 + B_2)) \quad (12)$$

When $x = \frac{-(F_2 - A_2 - K_1 - K_2 + K_3 + L_1 + L_2 + Q_1 + Q_2 + y(A_2 - F_1 - F_2))}{B_1 + B_2} = x^{**}$, $F(z) = 0$
and $d(F(z))/dz = 0$, which means the farmers cannot determine the stability strategy.

When $x \neq x^{**}$, the evolutionarily stable point may be $z = 0$ or $z = 1$. When $x > x^{**}$, $\frac{dF(z)}{dz}\Big|_{z=1} < 0$, $z = 1$ is the ESS of farmers. Conversely, $z = 0$ is ESS. The phase diagram of farmers' evolution is shown in Figure 2c.

### 4.2. Stability Analysis of Equilibrium Points

The evolutionary game algorithm shows that the stable state of the replicating dynamic system is affected by the initial probability of the players, and the initial probability of the players changes with time. When the replication dynamic equation is equal to 0, we can obtain the stable state of the system. By solving the dynamic Equations (3), (7) and (11), there are 15 equilibrium solutions.

However, as suggested in the literature [46], the stability points obtained by replication dynamic equations must be strictly in the Nash equilibrium of pure strategies, as other solutions show non-asymptotically stable states. Only the asymptotic stability of the eight special equilibrium solutions should be discussed, and can be determined as $X_1(0,0,0)$, $X_2(0,1,0)$, $X_3(0,0,1)$, $X_4(0,1,1)$, $X_5(1,0,0)$, $X_6(1,0,1)$, $X_7(1,1,0)$, and $X_8(1,1,1)$.

Based on the method of Friedman to judge the evolutionary equilibrium strategy [46], the stability of the differential equation can be analyzed by the Jacobian matrix, which is obtained from Equations (3), (7) and (11).

According to Lyapunov's stability theorem [76], when all the characteristic values ($\lambda$) of the Jacobian meet $\lambda < 0$, the equilibrium point is asymptotically stable. The equilibrium point is unstable when all the $\lambda$ of the Jacobian conform to $\lambda > 0$. Furthermore, if the $\lambda$ are mixed (some are positive and some are negative), the equilibrium point, also known as the saddle point (the conditional equilibrium that keeps the system stable only if the initial value changes), is unstable. By substituting the eight equilibrium points into the

Jacobian matrix, the λ of the Jacobian matrix corresponding to the equilibrium points can be obtained, as shown in Table 2.

$$
J = \begin{bmatrix} J_{11} & J_{12} & J_{13} \\ J_{21} & J_{22} & J_{23} \\ J_{31} & J_{32} & J_{33} \end{bmatrix} = \begin{bmatrix} \partial F(x)/\partial x & \partial F(x)/\partial y & \partial F(x)/\partial z \\ \partial F(y)/\partial x & \partial F(y)/\partial y & \partial F(y)/\partial z \\ \partial F(z)/\partial x & \partial F(z)/\partial y & \partial F(z)/\partial z \end{bmatrix}
$$

$$
= \begin{bmatrix} \begin{array}{c} (2x-1)(C_3 - B_2 + R - H_1 - H_2 + H_3 - V \\ +z(B_1 + B_2 - R + V - C_3) + y(C_2 - C_3) + 2C_3 y) \end{array} & x(x-1)(C_2 - C_3 + 2C_3 z) & x(x-1)(B_1 + B_2 - R + V - C_3 + 2C_3 y) \\ y(1-y)(C_2 - C_3 + 2C_3 z) & \begin{array}{c}(1-2y)(D_1 - F_2 - M_1 + M_3 + M_4 - T + N \\ +z(F_1 + F_2 - M_2 - M_4) + x(C_2 - C_3) + 2C_3 xz)\end{array} & y(1-y)(F_1 + F_2 - M_2 - M_4 + 2C_3 x) \\ z(1-z)(B_1 + B_2) & z(1-z)(A_1 + A_2 - F_1 - F_2) & \begin{array}{c}(1-2z)(F_2 - A_2 - K_1 - K_2 + K_3 + L_1 + L_2 \\ +Q_1 + Q_2 + y(A_1 + A_2 - F_1 - F_2) + x(B_1 + B_2))\end{array} \end{bmatrix} \quad (13)
$$

**Table 2.** Characteristic values in the Jacobian matrix.

| Equilibrium Points | $\lambda_1$ | $\lambda_2$ | $\lambda_3$ |
|---|---|---|---|
| $X_1(0,0,0)$ | $-(C_3 - B_2 + R - H_1 - H_2 + H_3 - V)$ | $D_1 - F_2 - M_1 + M_3 + M_4 - T + N$ | $F_2 - A_2 - K_1 - K_2 + K_3 + L_1 + L_2 + Q_1 + Q_2$ |
| $X_2(0,1,0)$ | $-(-B_2 + R - H_1 - H_2 + H_3 - V + C_2)$ | $-(D_1 - F_2 - M_1 + M_3 + M_4 - N + T)$ | $A_1 - K_1 - K_2 + K_3 + L_1 + L_2 + Q_1 + Q_2 - F_1$ |
| $X_3(0,0,1)$ | $-(B_1 - H_1 + H_3 - H_2)$ | $D_1 - M_1 + M_3 + N - T + F_1 - M_2$ | $-(F_2 - A_2 - K_1 - K_2 + K_3 + L_1 + L_2 + Q_1 + Q_2)$ |
| $X_4(0,1,1)$ | $-(C_2 - H_1 + H_3 + B_1 - H_2 + C_3)$ | $-(D_1 - M_1 + M_3 + N - T + F_1 - M_2)$ | $-(A_1 - K_1 - K_2 + K_3 + L_1 + L_2 + Q_1 + Q_2 - F_1)$ |
| $X_5(1,0,0)$ | $C_3 - B_2 + R - H_1 - H_2 + H_3 - V$ | $C_2 - C_3 + D_1 - F_2 - M_1 + M_3 + M_4 + N - T$ | $F_2 - A_2 - K_1 - K_2 + K_3 + L_1 + L_2 + Q_1 + Q_2 + B_1 + B_2$ |
| $X_6(1,0,1)$ | $B_1 - H_1 + H_3 - H_2$ | $D_1 - M_1 + M_3 + N - T + F_1 - M_2 + C_2 + C_3$ | $-(F_2 - A_2 - K_1 - K_2 + K_3 + L_1 + L_2 + Q_1 + Q_2 + B_1 + B_2)$ |
| $X_7(1,1,0)$ | $C_2 - B_2 + R - H_1 - H_2 + H_3 - V$ | $-(D_1 - F_2 - M_1 + M_3 + M_4 + N - T - C_3 + C_2)$ | $A_1 - K_1 - K_2 + K_3 + L_1 + L_2 + Q_1 + Q_2 - F_1 + B_1 + B_2$ |
| $X_8(1,1,1)$ | $C_2 - H_1 + H_3 + B_1 - H_2 + C_3$ | $-(D_1 - M_1 + M_3 + N - T + F_1 - M_2 + C_2 + C_3)$ | $-(A_1 - K_1 - K_2 + K_3 + L_1 + L_2 + Q_1 + Q_2 - F_1 + B_1 + B_2)$ |

Table 2 shows that although the evolutionary game analysis system gives a stable state under certain conditions, the stability of each Nash equilibrium cannot be determined with the known mathematical derivation method since many parameters are involved and the sign of the characteristic value depends on the value of the parameters. We need to evaluate by adding constraints.

Farmers' awareness and active behavior in protecting cultivated land are the main driving force to promote carbon neutrality. Therefore, $X_1$, $X_2$, $X_5$, and $X_7$ do not meet the ideal balance point of CLP. Only $X_3$, $X_4$, $X_6$ and $X_8$ are discussed below.

Case1: When the equilibrium condition meets $F_2 + K_3 + L_1 + L_2 + Q_1 + Q_2 > A_2 + K_1 + K_2$, $H_3 + B_1 > H_2 + H_1$, $D_1 + M_3 + N + F_1 < T + M_2 + M_1$, the ESS is $X_3(0,0,1)$. It means that local governments do not support CLP and focus on economic construction, while ATSOs tend to promote HCAT to maximize benefits. Although farmers have the initiative to protect farmland, it is unrealistic to rely on farmers' consciousness to complete low-carbon agricultural transformation when the government ignores it and lacks the support of LCAT. The balance point is not recommended.

Case2: When the equilibrium condition meets $A_1 + K_3 + L_1 + L_2 + Q_1 + Q_2 > K_1 + K_2 + F_1$, $C_2 + H_3 + B_1 + C_3 > H_2 + H_1$, $D_1 + M_3 + N + F_1 > T + M_2 + M_1$, the ESS is $X_4(0,1,1)$. The ATSOs chose to promote LCAT for farmers, which further strengthened farmers' willingness to take the initiative to adopt low-carbon production to protect farmland. In this case, the ATSOs and farmers have no loss of their interests and can rely on their consciousness to achieve farmland protection without governments' policy support and supervision. It is the suboptimal ESS.

Case3: When the equilibrium condition meets $D_1 + M_3 + N + F_1 + C_2 + C_3 < M_1 + T + M_2$, $F_2 + K_3 + L_1 + L_2 + Q_1 + Q_2 + B_1 + B_2 > A_2 + K_1 + K_2$, $B_1 + H_3 < H_1 + H_2$, the ESS is $X_6(1,0,1)$. This shows that the government can make up for farmers' losses in production through subsidies so that farmers take the initiative to apply for LCAT. However, the government's insufficient support for ATSOs leads to the lack of support and service for LCAT for farmers. Farmers have to grow their crops at a higher cost, which will increase the government's burden in the long run. It is not ideal for cash-strapped local governments.

Case4: When the equilibrium condition meets $D_1 + M_3 + N + F_1 + C_2 + C_3 > M_1 + T + M_2 C_2 + H_3 + B_1 + C_3 < H_2 + H_1$, $A_1 + K_3 + L_1 + L_2 + Q_1 + Q_2 + B_1 + B_2 > K_1 + K_2 + F_1$, the ESS is $X_8(1,1,1)$. In this case, the initiative of farmers to protect farmland will be supported by the government in terms of policies, while farmers will be supported by

ATSOs in LCAT so that farmers can persist in protecting farmland for a long time. The service demand of farmers is the driving force for the survival of ATSOs, and the subjects promoting LCAT get preferential policies from the government in the process of competition. Therefore, the interests of all stakeholders are ensured, and protection tillage technology will be steadily promoted to cover cultivated land in the main black soil areas by 2030. It is the optimal ESS.

Many factors affect the evolutionary stability strategy of the subject in the game. In this paper, the dynamic evolution process of local governments, ATSOs, and farmers' household behavior strategies are simulated by the simulation analysis method, and the evolutionary results are analyzed to provide a reasonable basis for the formulation of CLP policy.

## 5. Simulation Analysis and Discussion

As of 2021, Northeast China includes 121 county-level divisions in Heilongjiang Province, 100 in Liaoning Province, 60 in Jilin Province, and 36 banner counties in Inner Mongolia Autonomous Region, with a cultivated area of 278 million mu of typical blackland (mu, a Chinese unit of land measurement that equals to 1/15 of a hectare or 1/6 of an acre). In the *Outline of the Northeast Black Land Protection Plan (2017–2030)*, China aims to achieve 140 million mu of protected tillage area in the northeast by 2025 and 250 million mu by 2030, basically covering the cultivated land in the main black soil areas. Therefore, this paper mainly focuses on the typical blackland.

Based on the stability condition of the optimal equilibrium point $X_8$, this paper sets the initial values of the parameters according to the game relations and logical relations among the three stakeholders. In general, as long as the logical relationship between the parameters remains unchanged, the setting of initial parameters only affects the fluctuation amplitude of the curve, not the final convergence of the curve [77]. To make the simulation results close to reality and provide a reasonable basis for the formulation of black soil protection policies, this paper mainly sets the initial parameters based on the actual data in Northeast China. The data was obtained from the statistical yearbook, news reports, and relevant literature. Some difficult quantitative data were estimated by the survey of experts and managers. The initial assignment of the parameters is shown in Table A1

### 5.1. Evolution and Evaluation of the Effect of CLP Policy

In the theoretical analysis, we determined the corresponding conditions of stakeholder behavior strategy selection at different stages. This paper sets different initial strategy proportions to observe the evolution of the system as a whole.

From Figure 3, we can see that when the real situation of CLP satisfies the conditions of an ideal stable equilibrium, other equilibrium points may also exist because of the different initial strategy ratio values of stakeholders. When the initial strategy ratio of the ATSOs is higher than 0.5 and the initial strategy ratio of farmers is higher than 0.2, the game system will evolve to the optimal ESS (1,1,1); if the initial strategy ratio of the ATSOs is lower than 0.5, the game system will evolve and converge to the undesirable ESS (1,0,0). It can be found that the evolution result of the game model depends mainly on the initial strategy ratio of the ATSOs rather than the local governments and farmers. This is because local governments are called and supported by the national goal of carbon neutrality, and their attitudes toward CLP are positive in order to obtain political performance benefits. For farmers, with a strong awareness of CLP and a lack of support from ATSOs in LCAT, farmers find it hard to promote agricultural technology by their own power, which makes it difficult to insist on CLP for a long time, and they tend to continue to use HCAT instead, so the value of the farmers' initial strategy ratio is weak. In contrast, ATSOs can help farmers train and apply LCAT while strengthening their awareness of farmland protection, and their initial strategy ratio values dominate farmers' behavioral decisions, so the behavioral decisions of ATSOs become critical. Referring to Lanxi County, one of the pilot counties of China's agricultural socialized service innovation, its cultivated land area is 2.506 million mu in total, and the production trusteeship area of the ATSOs is planned to reach 1.2 million

mu. Assuming that 0.48 represents the initial strategy ratio value of the ATSOs, which is close to 0.5, it can be inferred that the CLP situation in the pilot area in Northeast China will break through the critical value and be the first to reach the ideal stability. Without changing the current farmland protection policy, we can assess that the pilot areas in Northeast China with good farmland protection publicity and the areas with poor farmland protection awareness will evolve into two situations in the future: full participation and only local government participation in farmland protection. In order to have full participation in CLP and reach the desired stability faster, we need to analyze the evolutionary trend of the system under different parameters.

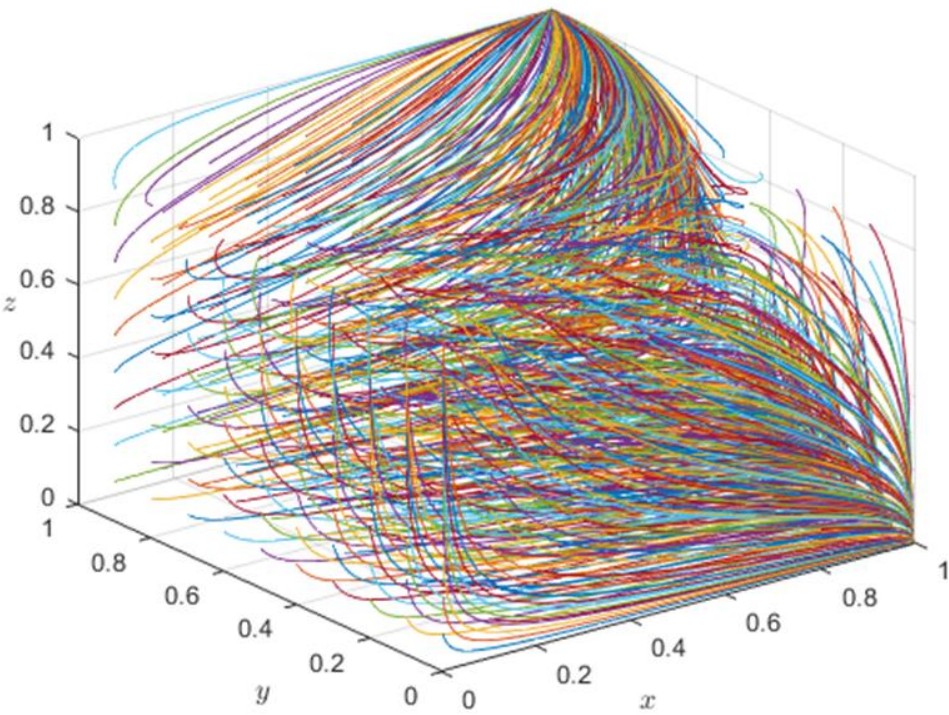

**Figure 3.** System evolution with different initial strategy ratios.

### 5.2. Sensitivity Analysis of Major Factors

We used sensitivity analysis to investigate how ESS and stakeholder strategies will be affected, focusing on the evolution of stakeholders with low initial strategy ratios and when ESS changes with high initial strategy ratios. Assume $(x_0, y_0, z_0)$ = (0.32, 0.48, 0.4) is the low initial strategy ratio—System A; $(x_1, y_1, z_1)$ = (0.7, 0.7, 0.7) is the high initial strategic ratio—System B.

### 5.2.1. Impact of Farmers' Subsidy $B_1$

Let $B_1$ take the values of 250, 350, 450, and 550, respectively, which are equivalent to raising farmers' subsidies from 100 to 140, 180, and 220 CNY/mu, respectively. From Figure 4a, after the increase of the farmer's subsidy to 140 CNY/mu, System A has gotten rid of the situation that only local government participates in CLP and increased the strategic proportion of farmers and ATSOs. However, with the subsidy amount reaching 220 CNY/mu, the regional cropland protection accelerates to the suboptimal ESS (0,1,1), and it is necessary to further analyze the evolutionary path of local governments in System A from Figure 5. When $B_1$ reaches 350, System A does maintain a brief stabilization point (1,1,1), but after 1.2 units of time, the strategic proportion of local governments that originally supported CLP gradually decreases. We can assume that the subsidy of 140 CNY/mu has increased the financial pressure on local governments. In particular, the strategic proportion of local governments decreases to zero within 0.1 unit of time when the

farmer subsidy is USD 220/mu, confirming that excessive subsidies lead to a weakening of local governments' willingness to participate in CLP.

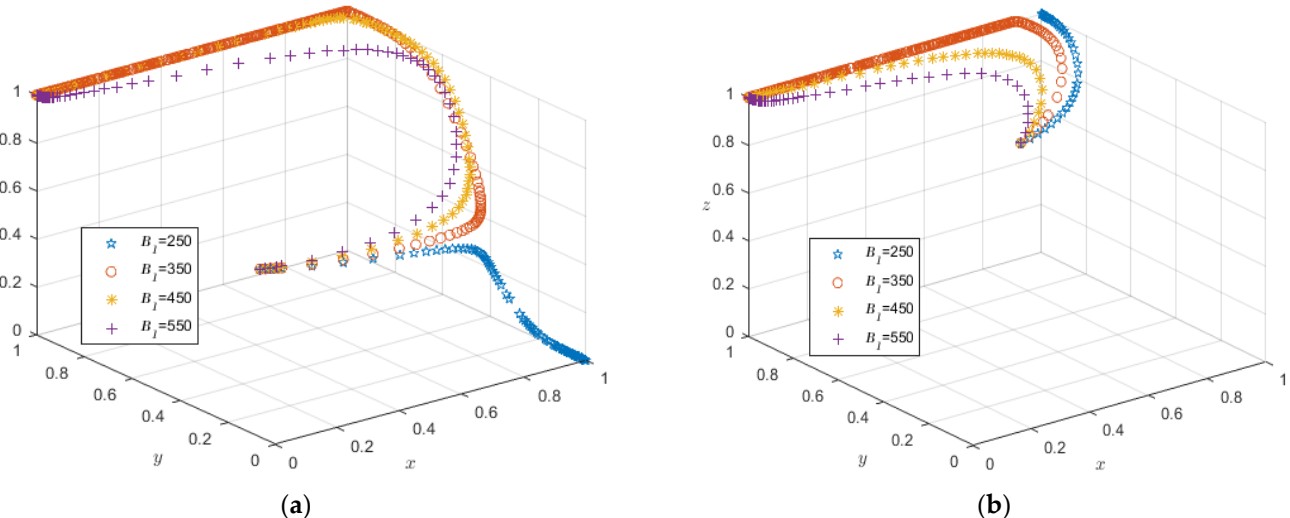

(**a**)　　　　　　　　　　　　　　　　　　　　　(**b**)

**Figure 4.** The evolution path of the different farmers' subsidies: (**a**) evolution path of System A with $(x_0, y_0, z_0) = (0.32, 0.48, 0.4)$; (**b**) evolution path of System B with $(x_1, y_1, z_1) = (0.7, 0.7, 0.7)$.

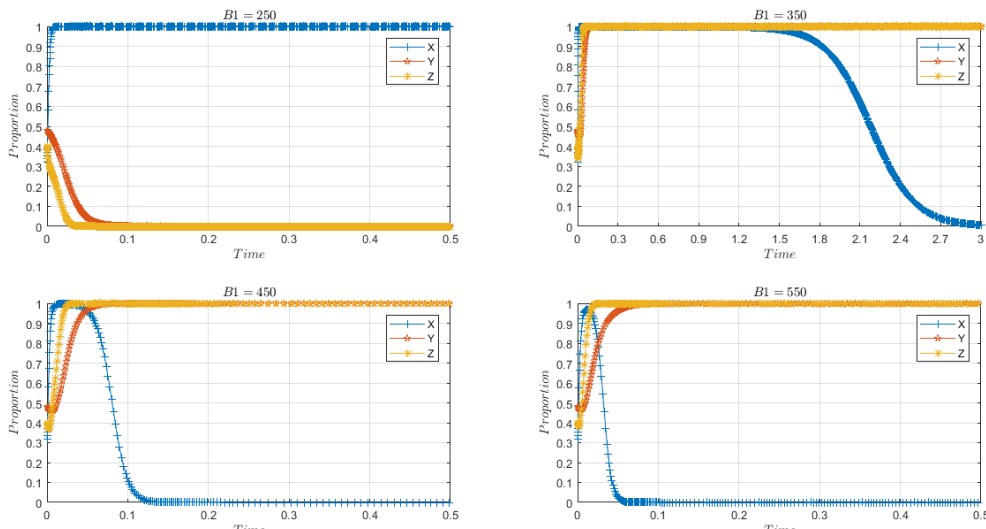

**Figure 5.** Evolution path of each stakeholder in System A under different farmers' subsidies.

Compared with System A, the initial strategic proportion of subjects in System B is high. The game model in Figure 4b has stabilized at the optimal ESS (1,1,1). When $B_1$ reaches 350, the ideal situation of full participation in System B is changed and finally converges to the sub-ideal stable point (0,1,1). Meanwhile, the increase in the farmer's subsidy accelerates the convergence of system B to (0,1,1). Therefore, it is not necessary to increase the farmer subsidy for areas with a high proportion of full participation in CLP.

### 5.2.2. Impact of Farmers' Fines $B_2$

According to the proposed increase of fines for the illegal sale of blackland in *The Draft Law on the Protection of Black Land in 2022*, we supposed $B_2$ from 21.1 to 100, 200, and 300.

From Figure 6, the increase in farmers' fines can make System A evolve from (1,0,0) to the optimal ESS, (1,1,1), and also accelerate the evolution trend of System B to the optimal ESS. Results indicate that the increase in farmers' fines can be implemented as a long-term CLP policy in Northeast China regardless of the proportion of the initial strategy.

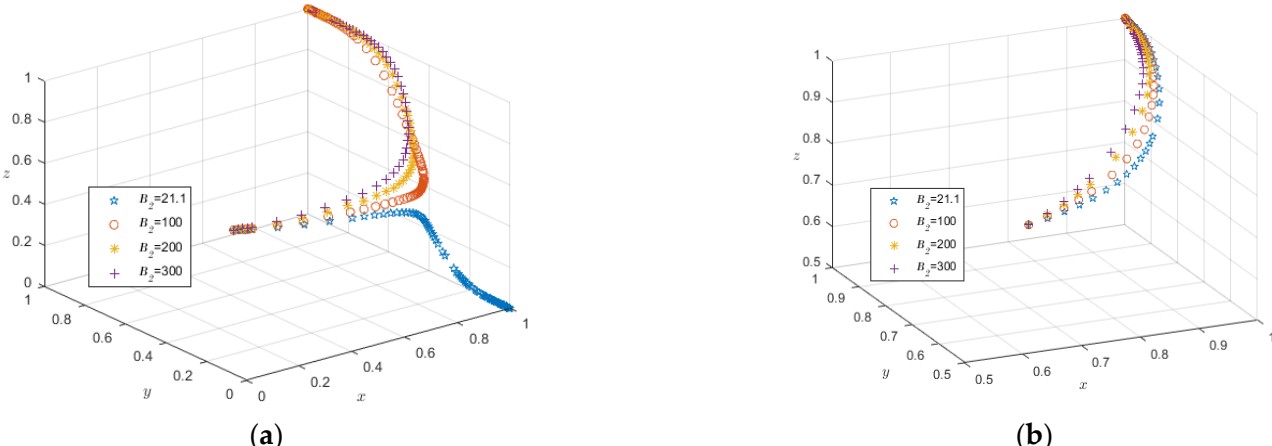

**Figure 6.** The evolution path under different farmers' fines: (**a**) evolution path of System A with $(x_0, y_0, z_0) = (0.32, 0.48, 0.4)$; (**b**) evolution path of System B with $(x_1, y_1, z_1) = (0.7, 0.7, 0.7)$.

In Figure 7, when $B_2$ is increased to 100, farmers have a "hesitation" time and do not immediately choose "not protect". This is because higher fines force farmers to bear the cost of the crime, and there is time to decide whether to protect their farmland. When $B_2$ continues to increase to 200 and 300, farmers evolve more quickly and stabilize to the strategy of "protect" due to the deterrent of fines, and ATSOs also evolve to the "LCAT" strategy with the influence of farmers. It shows that the increase in fines has an inhibitory effect on farmers' illegal digging of black soil.

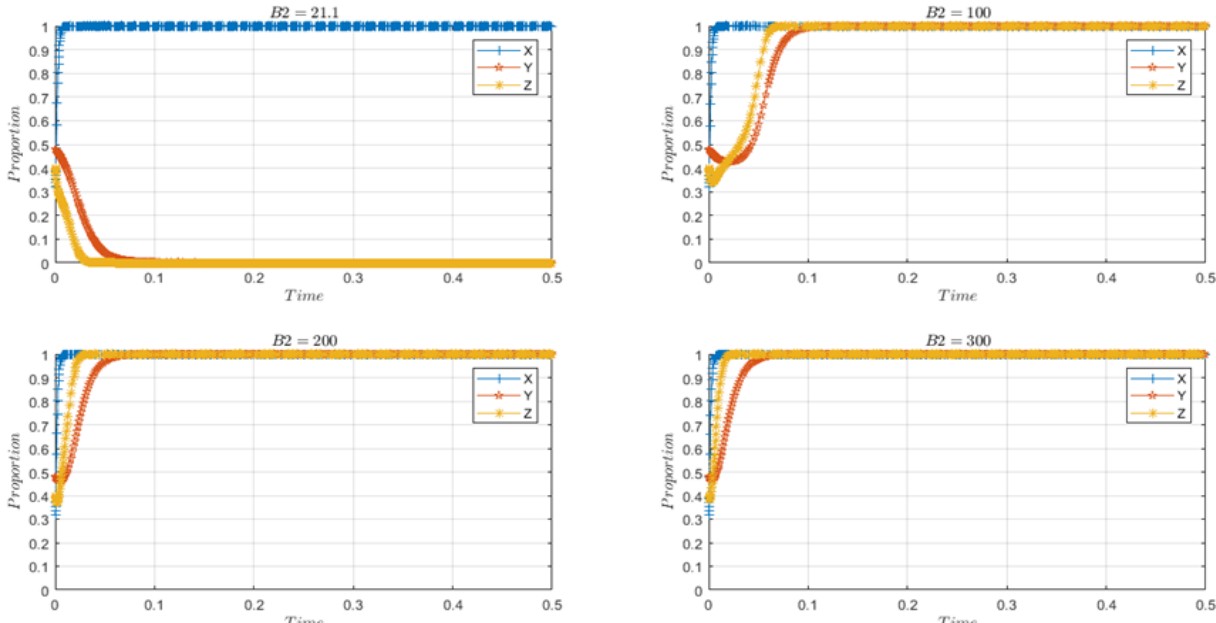

**Figure 7.** Evolution path of each stakeholder in System A under different farmers' fines.

### 5.2.3. Impact of Incentive Dedicated Funds $C_2$ and Production Trusteeship Subsidies $C_3$

At present, the government strongly supports the development of ATSOs and encourages socialized service organizations to provide production trusteeship services, to analyze and compare how $C_2$ and $C_3$ can better motivate ATSOs to apply LCAT in service. We assume that $C_2$ increases from 18 to 50,100,150, and $C_3$ from 11.89 to 70,120,150, respectively.

From Figure 8a,c, when $C_2$ increases to 100, it is beneficial for System A to evolve to the optimal ESS, but when $C_2$ reaches 150, it makes System A evolve to the suboptimal ESS. On the contrary, the increase in $C_3$ accelerates the evolution of System A to the stable point

(1,0,0), which is not helpful for the promotion of LCAT. This is because the production trusteeship subsidies issued by local governments do not distinguish whether the ATSOs apply LCAT. From Figure 8b,d, the increase in $C_2$ and $C_3$ will make System B gradually deviate from the optimal stability point, and finally converge to the suboptimal ESS (0,1,1), indicating that the subsidy amount of local government should be controlled within a certain range. It can be seen from Figure 8 that local governments setting appropriate incentive funds can maximize the incentive for ATSOs to promote LCAT. However, no matter how much they improve the production trusteeship subsidies, it has no effect.

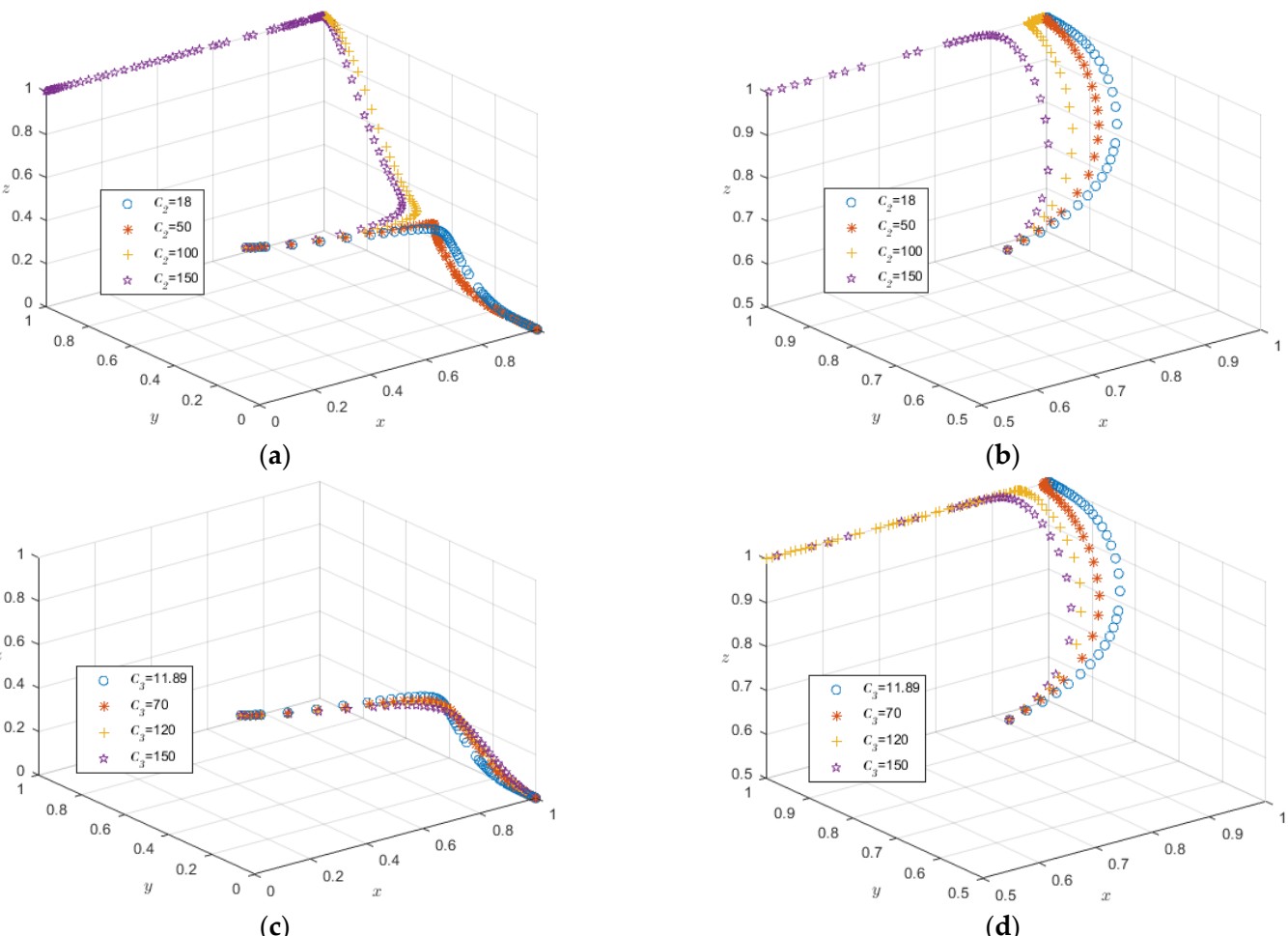

**Figure 8.** The evolution path under different incentive dedicated funds: (**a**) System A with $(x_0, y_0, z_0)$ = (0.32, 0.48, 0.4); (**b**) System B with $(x_1, y_1, z_1)$ = (0.7, 0.7, 0.7). The evolution path under different production trusteeship subsidies: (**c**) System A with $(x_0, y_0, z_0)$ = (0.32, 0.48, 0.4); (**d**) System B with $(x_1, y_1, z_1)$ = (0.7, 0.7, 0.7).

### 5.2.4. Impact of the Production Trusteeship Charge of LCAT $F_1$ and Low Carbon Production Trusteeship Service Cost $M_2$

The low carbon production trusteeship service is the main factor affecting farmers to protect farmland. ATSOs with the goal of carbon neutrality need to pay higher costs to promote LCAT, which also leads to higher service fees charged to farmers, thus affecting whether farmers choose the low carbon production trusteeship services. To research which methods can be adopted by the ATSOs to better obtain competitiveness and make the system converge to the ideal state, four methods can be assumed: reduce charges and costs ($F_1$ = 900, $M_2$ = 770); reduce costs only ($F_1$ = 1000, $M_2$ = 770); do not reduce charges and costs ($F_1$ = 1000, $M_2$ = 925); and raise charges and costs ($F_1$ = 1200, $M_2$ = 925).

Figure 9 shows that System A of the ATSOs help converge to (1,1,1) by "reduce charges and costs" or "reduce costs ", but the method of "raise charges and costs" is not effective. For System B, all four methods will not change the steady state of its optimal ESS, which is (1,1,1). It can be seen that the ATSOs in each region can adopt the method of "reducing costs" to promote the overall convergence of the system to the optimal result.

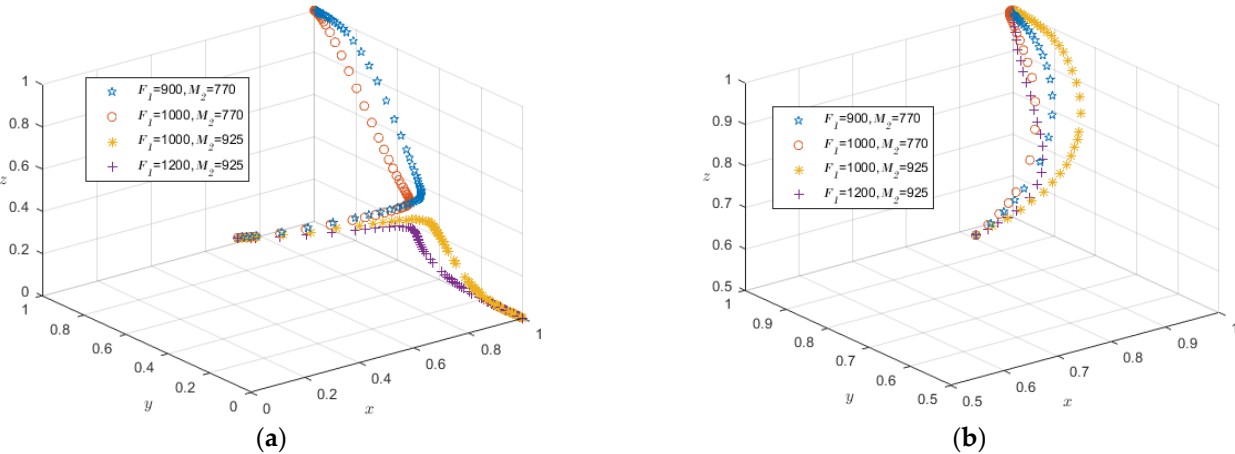

(**a**)　　　　　　　　　　　　　　　(**b**)

**Figure 9.** Evolution path of $F_1$ and $M_2$ under different values: (**a**) System A with $(x_0, y_0, z_0)$ = (0.32, 0.48, 0.4); (**b**) System B with $(x_1, y_1, z_1)$ = (0.7, 0.7, 0.7).

Figure 10 shows that the convergence rate of ATSOs and farmers is faster when $F_1$ = 1000, reaching a steady state in 0.04 units time, compared with "reduce charges and costs" and "reduce costs". The reason is that ATSOs can gain more profits by reducing costs and having strong competitiveness. They can take the initiative to expand the scale of serving farmers and thus improve farmers' awareness of protecting farmland. Therefore, it is the most effective method for the ATSOs to promote LCAT by only reducing the low carbon production trusteeship service costs, specifically the cost of service to 308 CNY/mu (770/2.5 = 308).

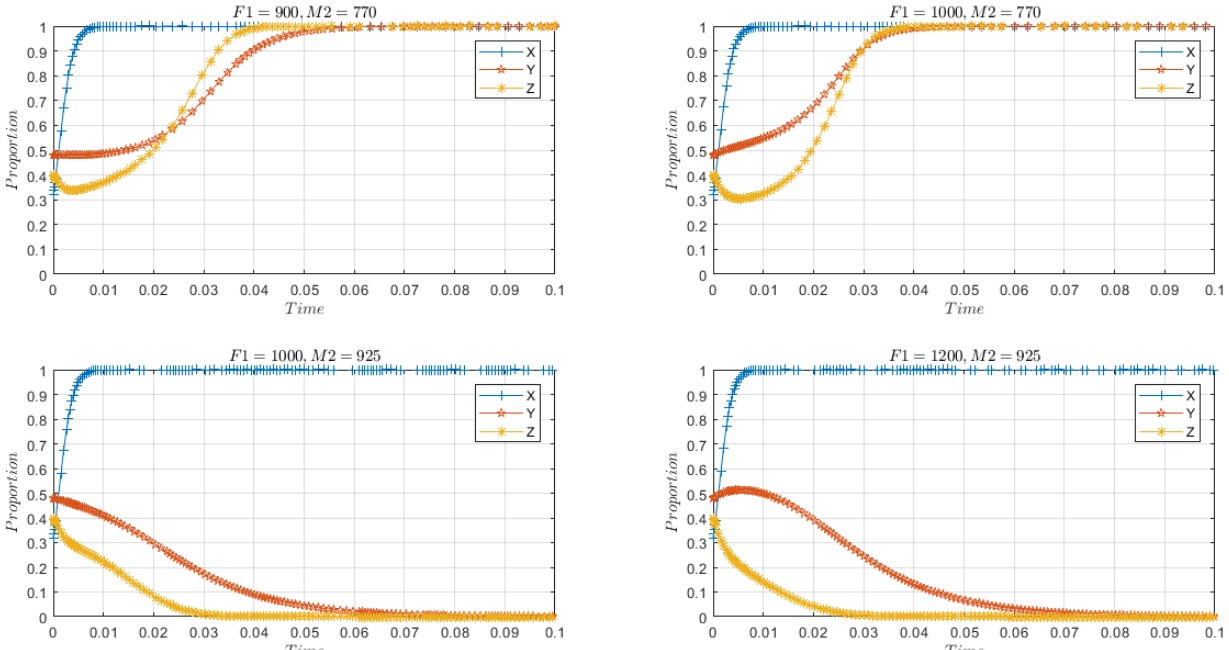

**Figure 10.** Evolution path of each stakeholder in System A under the influence of $F_1$ and $M_2$.

## 6. Discussion

CLP involves conflicting interests among multiple subjects. The behavioral decisions of each subject directly or indirectly affect CLP. Therefore, it is worth thinking about how to coordinate the interests of each subject and how to adjust measures to improve the overall effect of arable land protection. There is existing fallow land protection, with the aim of improving the productivity of cropland for stakeholders [58], and studies on abandonment management [59], but studies related to CLP are still not abundant. As countries pay more and more attention to carbon neutrality, a low-carbon, efficient CLP becomes an urgent need. Therefore, this paper is a further innovation in CLP by studying the behavioral decisions of subjects in CLP from the perspective of agronomic extension under the context of carbon neutrality. This paper constructs an evolutionary game model, including farmers, ATSOs, and local governments, and explores the decision-making mechanism of the interaction among the subjects, analyzes the evolutionary process of ATSOs in CLP, and extends the scope of research on subjects in ATSOs.

In Section 5.1, we analyzed the system evolution trend under different initial ratio values and assessed that there are two future outcomes of CLP in Northeast China after satisfying the optimal equilibrium condition: full participation and local government participation only. The *Northeast Blackland Protection Plan (2017–2030)* issued by China requires local governments to support CLP and sets up project funding to alleviate the financial pressure on local governments, which enables local governments to implement the policy well. Therefore, regardless of the initial proportion, local governments can adhere to support CLP. The initial proportion value reflects the willingness of subjects to participate in CLP. Under the initial conditions, areas with an initial ratio lower than 0.5 can be considered as having a backward awareness of CLP, which leads to the fact that only the local government participates in CLP in some remote areas with weak propaganda, while other subjects do not pay attention to it. On the contrary, the pilot areas with good publicity of CLP can form a situation where all subjects can participate. This helps us to understand the situation of CLP in Northeast China. The government's work is focused on ideological propaganda and cultivation, and it is necessary to combine the efforts of non-governmental organizations to raise awareness of all subjects on CLP [78].

According to the results of the game, adjusting the subsidies and fines of farmers can be a key method to improve the overall effect of CLP. Local governments with a low initial strategy ratio can achieve the ideal equilibrium by increasing farmers' subsidies, but the willingness of local governments to participate in CLP decreases when subsidies are over 140 CNY/mu. The reason is that too high subsidies impose a greater financial burden on local governments. This is consistent with the finding by Sibande, in that an appropriate agricultural subsidy program helps to increase the participation of smallholders [79]. We further found that the ideal equilibrium has been achieved in areas with a high proportion of initial strategies, and increasing farm subsidies has negative effects, so local governments need to develop a reasonable farm subsidy system according to the regional situation. In order to control the amount of subsidy, local governments can provide incentives for farmers to reduce the additional carbon emissions generated by self-growing by actively adopting the production escrow system. Or they can order a unified production trust service, and regulate the agricultural production with unified seeds, pesticides, fertilizers, and management practices. In addition, the game model does not deviate from the optimal ESS with the increase of fines because the cost of not protecting farmland is higher than the cost of protecting farmland. Therefore, the increase in farmers' fines can be a long-term farmland protection policy. We suggest improving the black land protection law, strictly catching the phenomenon of black soil poaching, increasing the reward for public reporting, reducing the cost of public reporting by various means, and actively promoting and developing the public supervision mechanism.

A similar trend is observed for the incentive dedicated funds given by local governments to ATSOs compared to farmer subsidies. The game model converges to the optimal ESS when the incentive funds reach 5 billion CNY. If beyond this value, the local

government will lose its enthusiasm to support CLP due to the high financial expenditure. Therefore, the incentive dedicated funds need to be maintained at an appropriate range. However, no matter how the local government increases the production trust subsidy, it will not have a favorable impact on the ATSOs. The reason is that local governments do not evaluate the ATSOs, and do not screen whether the agricultural technologies promoted by them are beneficial to farmland protection. The ATSOs that do not apply LCAT can also enjoy subsidies, so the ATSOs that promote LCAT do not have the advantage of financial support. In this regard, we can suggest the government provide technical support, preferential policies, and other ways to guide ATSOs to promote LCAT in production trusteeship.

From the perspective of ATSOs, "reduce costs only" is the most effective way to motivate the main ATSOs to promote LCAT, and the service fee is 400 CNY/mu and the service cost is reduced to 308 CNY/mu. In order to expand the benefits and gain competitiveness, the ATSOs promoting LCAT need to keep the service fee amount unchanged to maintain the farmers' group and reduce the service cost to prepare for the expansion of farmers' service scale. Therefore, it is suggested that (1) ATSOs should give full play to advanced agricultural machinery and equipment to improve production efficiency and reduce production operation costs; centralize the procurement of organic agricultural materials and promote soil testing and fertilization to reduce physical and chemical costs; and adopt new varieties and implement standardized production to improve the quality of farm products and achieve cost savings and efficiency gains. (2) Regularly organize training techniques for agricultural technicians, such as less tillage and no-till cultivation, straw return technology, etc.; develop a relative assessment system; and actively attract high-level talents. (3) Actively and proactively go to the countryside to improve farmers' understanding of LCAT and raise their awareness of CLP, thus expanding the organization's influence and enhancing its competitiveness, ultimately continuously improving the contribution of CLP in carbon neutrality.

## 7. Conclusions

In the context of carbon neutrality, this study constructs an evolutionary game model from the perspective of agricultural extension, focusing on CLP; analyzes the evolutionary process of farmers, ATSOs, and local governments; and explores the decision-making mechanism of the interactions among them. By comparing the effects of external variables on the strategies of participants under different conditions, we propose a reasonable policy to coordinate the conflicts and make the game converge to the ideal equilibrium strategy combination desired by the society, so as to promote arable land protection and enhance the carbon-neutrality effect. The following conclusions were drawn:

(1) From the main body's balance analysis and evolutionary stability analysis, it can be seen that the local governments, ATSOs, and farmers are affected by the strategies of the other two stakeholders. Among the eight pure strategy points obtained by replicating the dynamic equation, $X_6(0,1,1)$ is the suboptimal ESS, $X_8 (1,1,1)$ is the optimal ESS, and the other points do not conform to the ideal equilibrium point of the system. Three constraint conditions are required for the model to reach the ideal equilibrium point $X_8$: $D_1 + M_3 + N + F_1 + C_2 + C_3 > M_1 + T + M_2$, $A_1 + K_3 + L_1 + L_2 + Q_1 + Q_2 + B_1 + B_2 > K_1 + K_2 + F_1$, $C_2 + H_3 + B_1 + C_3 < H_2 + H_1$.

(2) From the numerical simulation analysis, it can be seen that due to the different initial probability strategy ratios of stakeholders, there are two future outcomes of CLP in Northeast China after satisfying the optimal equilibrium condition: full participation (1,1,1) and local government participation only (1,0,0). The evolution of the game model depends mainly on the initial strategy ratio of the ATSOs. When the value of the initial strategy ratio of ATSOs is lower than 0.5, it can be considered as an area where the main body is backward in awareness of CLP. From further sensitivity analysis, the following conclusions can be drawn:

(1)   Appropriate farmer subsidies can help to carry out arable land protection, which should be 100~140 CNY/mu, and if it is more than 140 CNY/mu, it will lead to the weakening of local government's willingness to participate in CLP. Therefore, the government's financial burden should be reduced by eliminating the excessive subsidies to farmers.

(2)   An increase in the amount of the fines has a disincentive effect on farmers digging black soil, and the game model achieves an ideal equilibrium when it reaches 10 billion CNY. An effective penalty can make the government achieve its desired goal with fewer subsidies, which can be implemented as a long-term farmland protection policy.

(3)   Adequate incentive funds can provide the greatest incentive for ATSOs to promote LCAT, but note that this should be maintained at 5 billion CNY, otherwise it will have a negative impact on local governments. In addition, the production trusteeship subsidy provided by the government has no favorable impact on the ATSOs, which should be replaced by other ways, such as technical support and preferential policies, for guidance.

(4)   The main body of ATSOs adopts the method of reducing the service cost of production trusteeship and not raising the service fee, the service fee being maintained at 400 RMB/mu, and the service cost being reduced to 308 RMB/mu, which can most effectively increase the competitiveness of the main body of agrotechnology services in promoting LCAT.

With the rise of the power of socialized service organizations in agricultural service organizations, their services are not only limited to agricultural extension but also include services such as pre-production purchase and pre-production sales; this paper only explores the CLP in the mid-production link by ATSOs, thus failing to cover the whole process. In the future, this paper will consider the more complex decision-making process between related agricultural materials, agricultural products, and other industries, farmers, and agricultural service providers. At the same time, the development of new models, such as carbon tax and carbon trading in agriculture, and their impacts on stakeholders, will also be studied as the process of carbon neutrality accelerates.

**Author Contributions:** Conceptualization, Y.T.; methodology, Y.T.; software, P.L.; validation, P.L.; formal analysis, P.L.; investigation, Y.T. and P.L.; resources, P.L.; data curation, P.L.; writing—original draft preparation, Y.T. and P.L.; writing—review and editing, Y.T. and P.L.; visualization, P.L.; supervision, Y.T.; project administration, Y.T.; funding acquisition, Y.T. All authors have read and agreed to the published version of the manuscript.

**Funding:** This research was funded by The National Social Science Fund of China, the subject name is Study on the Multi-driven Mechanism of Northeast Blackland Conservation Based on Total Quality Management (No. 22BJY240).

**Institutional Review Board Statement:** Not applicable.

**Informed Consent Statement:** Not applicable.

**Data Availability Statement:** Details regarding where data supporting the article can be found in the following websites: http://www.moa.gov.cn/xw/zwdt/202012/t20201218_6358374.htm, http://agri.jl.gov.cn/xdny/nykj/kycg/202103/t20210331_7983335.html, http://www.datajci.com, http://www.hljyian.gov.cn/pages/5edef7791d41c809aa235fa2, https://www.hljlanxi.gov.cn/news/5014.html, http://agri.jl.gov.cn/xwfb/sxyw/202107/t20210727_8153628.html, https://www.huoqiu.gov.cn/public/6618291/31221191.html.

**Acknowledgments:** The authors would like to thank the reviewers and the editor, whose suggestions greatly improved the manuscript.

**Conflicts of Interest:** The authors declare that they have no known competing financial interests or personal relationships that could have appeared to influence the work reported in this paper.

# Appendix A

**Table A1.** Initial assignment of the variable parameters based on real data.

| Parameters | Meaning | The Source of Data | Data |
|---|---|---|---|
| $C_1$ | The subsidy for reform and construction of the agricultural technology extension system | In 2015, the subsidy was 2.6 billion CNY in China, covering 1200 agricultural counties nationwide. The annual growth rate is 25%. It is estimated that in northeast China, it is $26 \times (121 + 100 + 60 + 36) \times (1.25)^6/1200 = 2.707$ billion CNY. | 27.07 |
| $C_2$ | The incentive dedicated funds set up by local governments for ATSOs promoting LCAT | In 2022, Liaoning province will raise 620 million CNY of funds for protection tillage and blackland protection to support the promotion of advanced applicable technologies of protection tillage, which can be estimated to be 1.8 billion CNY in total. | 18 |
| $C_3$ | The subsidies for production trusteeship of the ATSOs | From 2017 to 2019, the subsidies for agricultural production trusteeship are 11 billion CNY, and the subsidies will increase to 4.5 billion CNY in 2022. It is estimated that $45 \times (121 + 100 + 60 + 36)/1200 = 1.189$ billion CNY. | 11.89 |
| $B_1$ | The subsidy is given to farmers who protect cultivated land | According to the Notice of Liaoning Province on the Implementation Plan of Black soil Protection Tillage in 2022, the subsidy standard is 58 CNY/mu. The subsidy standard of Heilongjiang Province and Jilin Province is 57 CNY/mu and 40 CNY/mu respectively. The average subsidy is 52 CNY/mu, and we can estimate $52 \times 250 = 13$ billion CNY. Other subsidies, such as subsoiling and organic fertilizer subsidies, are 12 billion CNY. | 250 |
| $B_2$ | The fine for farmers who do not protect their cultivated land | Due to the illegal occupation of a small number of fines, the main calculation is stolen digging black soil fines. In 2017, 200,000 cubic meters of black soil were stolen and dug in Jilin Province, and the fine for illegally selling black soil was 2000 $CNY/m^3$, which can be estimated as 400 million CNY. It is estimated that $4/60 * \times (121 + 100 + 60 + 36) = 2.11$ billion CNY. | 21.1 |
| $R$ | The cost of governance and publicity for the deterioration of the cultivated land environment | Refer to the central blackland protection fund of 80 million CNY issued by the Hulunbuir Finance Bureau in 2022 to support the work related to blackland protection in northeast China. We can estimate $8000 \times (4 + 8 + 14 + 13) = 3.12$ billion CNY. | 31.2 |
| $H_1$ | The performance benefits and social benefits brought by the increase in grain yield | In 2021, the incentive fund for grain-producing counties from Liaoning, Jilin, Heilongjiang, and Inner Mongolia provinces is 1.14, 0.5, 4.312, and 1.42 billion CNY respectively. Total of 7.38 billion CNY. | 73.8 |
| $H_2$ | The long-term comprehensive benefits of governments from carbon neutrality | Long-term comprehensive benefits are difficult to estimate based on existing data and are mainly estimated through expert surveys. | 600 |
| $H_3$ | The economic construction benefits of local governments | Estimated by expert survey method. | 300 |
| $V$ | the long-term benefit loss caused by local governments' inaction | According to the Ministry of Agriculture and Rural Affairs in China, the annual loss of black soil leads to a decrease in grain production of 20 million tons. Taking corn, the main food crop, as a reference, the average price is 2670 CNY/ton, and the estimated grain loss is 53.4 billion CNY. The growth of construction land brought about 9.98 million tons of carbon emissions from 2000 to 2015. The social cost of carbon dioxide emissions in China is 24 USD/ton [80]. $998 \times 24 \times 67,000$ CNY = 1.6 billion CNY. The long-term benefit loss should be over 55 billion CNY. We can estimate 70 billion CNY. | 700 |
| $C_4$ | Working and operating expenses | Taking the Agricultural Technology Extension Center of Yi'an County in Heilongjiang Province as a reference, the funds for the daily operation of the organs were 937,700 CNY and 488,900 CNY respectively. As of 2016, there are 1555 ATSOs in Heilongjiang Province, 1111 ATSOs in Jilin Province, and 5620 ATSOs in Liaoning Province, with a total of 8286. According to the average estimate of work funds, $(93.77 + 48.89)/2 \times 8286 = 5.91$ billion CNY. | 59.1 |

**Table A1.** *Cont.*

| Parameters | Meaning | The Source of Data | Data |
|---|---|---|---|
| $T$ | The capacity building costs for agricultural technicians | According to the Implementation Plan of Subsidy Project for The Reform and Construction of Grassroots Agricultural Technology Extension System in Huoqiu County, Zhejiang Province in 2020, the talent training amount is 569,000 CNY. We estimated $8286 \times 569,000$ CNY = 4.71 billion CNY. | 47.1 |
| $M_1$ | The expenditures related to the demonstration and promotion of LCAT in villages | Related expenses are mainly through manager survey. | 500 |
| $M_2$ | The costs of ATSOs to provide farmers with low carbon production trusteeship service | The profit of ATSOs' service is about 30 CNY per mu. Combined with the production trusteeship fee of LCAT, we can estimate the total cost is $2.5 \times (400 - 30) = 92.5$ billion CNY. | 925 |
| $M_4$ | The costs of ATSOs to provide farmers with conventional production trusteeship service | Combined with the production trusteeship fee of HCAT, we can estimate the total cost is $2.5 \times (338 - 30) = 77$ billion CNY. | 770 |
| $F_2$ | The production trusteeship fee of HCAT charged to farmers | Taking corn planting in Lanxi County, Heilongjiang province as a reference, the trusteeship service organization can reduce the planting cost by 10–15%, save labor costs by more than 40%, and reduce fertilizer application by more than 20% compared with farmers' self-planting. The estimated trusteeship cost is 338 CNY/mu, a total of $2.5 \times 3.38$ million = 84.5 billion CNY. | 845 |
| $F_1$ | The production trusteeship fee of LCAT charged to farmers | ATSOs adopt LCAT needs a higher cost. The estimated cost is 400 CNY/mu. Production trusteeship costs are $250 \times 40$ billion CNY =100 billion CNY. | 1000 |
| $D_1$ | The long-term development benefit of ATSOs due to the carbon-neutrality effect | It is difficult to estimate long-term comprehensive benefits based on existing data, so it is mainly estimated by experts. | 200 |
| $M_3$ | The related expenditure for conventional agricultural technology popularization | Related expenses are mainly through manager survey. | 100 |
| $N$ | The adverse impact of the promotion of HCAT on long-term development | Adverse effects are mainly estimated by experts. | 240 |
| $A_1$ | The production cost of self-planting with HCAT | Taking the protective tillage task area of 80 million mu in 2022 as the calculation area and corn as the example, farmers' self-planting cost of conventional agricultural production is 480 CNY/mu (including fertilizer costs of 150 CNY/mu, labor and agricultural machinery costs 280 CNY/mu, pesticides, and other costs 50 CNY/mu). Assuming no land rent cost, $2.5 \times 480 = 120$ billion CNY. | 1200 |
| $K_1$ | Farmers' crop benefits caused by yield increase technology of HCAT | The yield of corn in northeast China is calculated according to the high yield of 750 kg/mu, the average purchase price of corn is 0.6 CNY/kg, the corn planting subsidy is 68 CNY/mu, the fertilizer subsidy is 11 CNY/mu, minus the land rental cost of 500 CNY/mu, then the total income is about 1379 CNY/mu, $1379 \times 2.5 = 344.75$ billion CNY. | 3447.5 |
| $K_2$ | The illegal farmers' profits from the digging of black soil | In 2017, about 200,000 cubic meters of black soil were stolen from several regions in Jilin Province, involving more than 10 million CNY. It is estimated that the total illegal income is $1000 \times (121 + 100 + 60 + 36) = 3.17$ billion CNY. | 31.7 |
| $Q_1$ | The loss of farmland fertility caused by long-term predatory production | According to the statistics of the Ministry of Agriculture and Rural Affairs, the amount of nitrogen, phosphorus, potassium and other minerals lost in the soil in the northeast black soil area every year amounts to 4 million to 5 million tons when converted into standard fertilizers. According to the fertilizer costs 2800 CNY/ton, it is estimated that $450 \times 2800 = 12.6$ billion CNY. | 126 |
| $Q_2$ | The adverse impact of the increase in carbon emissions caused by HCAT | Estimated by expert survey method. | 200 |

Table A1. *Cont.*

| Parameters | Meaning | The Source of Data | Data |
|---|---|---|---|
| $A_2$ | The protective production cost paid by farmers for self-planting with LCAT | The cost of organic fertilizer application is 230 CNY/mu, the cost of pesticide reduction is 40 CNY/mu, the cost of technical training and equipment is 300 CNY/mu, and the estimated self-planting cost is about 570 CNY/mu. Excluding the cost of land lease, the total production cost is $2.5 \times 570 = 142.5$ billion CNY. | 1425 |
| $K_3$ | The crop income earned by farmers from self-planting with LCAT | According to the calculation of the average 600 kg/ mu, the total income is about 1058 CNY/mu, which can be $1008 \times 2.5 = 252$ billion CNY. | 2520 |
| $L_1$ | The increased potential of future crop yield by low carbon production trusteeship | To implement 250 million mu of protected blackland by 2030, farmers will adopt whole-process trusted-production, including labor costs, and adopt the protection tillage mode of trusted-production, saving 100 CNY and increasing efficiency per mu of corn. It can be assumed that the potential increase in crop income in the future is $100 \times 2.5 = 25$ billion CNY. | 250 |
| $L_2$ | Carbon-neutrality benefits for farmers | Carbon-neutrality benefits are difficult to estimate based on existing data, so they are mainly estimated by experts. | 300 |

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
