# Peer review of "Research on Behavioral Decision-Making of Subjects on Cultivated Land Conservation under the Goal of Carbon Neutrality"

_land, doi:10.3390/land11101664_

Round 1
Reviewer 1 Report
Manuscript entitled “Research on behavioral decision-making of subjects on cultivated land conservation under the goal of carbon neutrality” by Teng and team.
Authors highlighted the importance of protected cultivation to mitigation climate change and food safety and carbon neutrality.
Abstract: quantify data is missing
Keywords: Ok
Introduction: written well, in end of the introduction pl specific the objectives of this study or targets.
Game Model: Well define,
Simulation analysis and discussion: Written well, following modification is suggested
Evolution and evaluation of the effect of cultivated land conservation policy: quantify data is missing
Sensitivity analysis of major factors; OK
Impact of farmers’ subsidy B1: add quantify data and discussion is missing
Figure – OK
Discussion-: authors reported results only and discussion is missing (need major modification)
Conclusions -: quantify data is missing
Suggestion: Ok
Overall, the manuscript has very information, but discussion is missing.
Abstract and conclusion should be very specific its look like very general statements, data should be quantify.
Author Response
Dear Reviewer:
Those comments are all valuable and very helpful for revising and improving our paper, as well as the important guiding significance to our researches. Revised portion are marked in yellow in the revised manuscript. Please see the attachment
The main corrections in the paper and the responds to the reviewer’s comments are as flowing:
- Comment: Abstract: quantify data is missing
Response: In lines 11-34, we have reconstructed the abstract, adding quantitative data to make it more concrete.
- Comment: Introduction: written well, in end of the introduction specific the objectives of this study or targets.
Response: In lines 111-120, We highlight the aims at the end of the introduction, which aim to analyze the behavioral decisions of each subject, and identify the key issues of interest game focus and CLP, then consider how to improve government policies.
- Comment: Evolution and evaluation of the effect of cultivated land conservation policy: quantify data is missing.
Response: In lines 509-536, we have revised the Evolution and evaluation of the effect of cultivated land conservation policy, added quantitative data, and optimized the textual presentation.
- Comment: Impact of farmers’ subsidy B1: add quantify data and discussion is missing
Response: In lines 548-567,we have revised this section to add quantitative data, as well as a full discussion around data and game images.
- Comment: Discussion-: authors reported results only and discussion is missing (need major modification)
Response: In lines 635-715, we have revised the discussion section in detail as a separate part of the chapter. Our discussion is enriched by research comparisons, analysis of findings, and recommendations for countermeasures.
- Comment: Conclusions -: quantify data is missing
Response: In lines 717-767, we revised the conclusion and optimized the conclusion by combining the quantify data.
We appreciate for Reviewers’ warm work earnestly, and hope that the correction will meet with approval.

Reviewer 2 Report
I congratulate the authors for their article entitled "Research on behavioral decision-making of subjects on Cultivated land conservation under the goal of carbon neutrality" which constructs an evolutionary game model among local governments, agricultural technology service organizations (ATSOs), and farmers to analyze the conflicting interests and decision-making behavior of stakeholders.
Even though I find the overall article to be fluent and well developed, more information on the following sections should be added/modified:
*The Introduction section should be split into 2 - Introduction and Literature Review; Moreover, the Literature review should aim to present international studies from your field of study as most of your literature is from China;
* In the Game model part, there should be 1-2 paragraphs dedicated to presenting similar papers that apply the same or similar model in the same field or a different one;
* In the Conclusions section I would add some limitations of this study and further research gaps to be tackled;
* Fine tuning is needed throughout the entire article - for example, line 453 and line 513 - Figure 3 & 6 (with capital F); line 605 - Suggestions (capital S).
Author Response
Dear Reviewer:
Those comments are all valuable and very helpful for revising and improving our paper, as well as the important guiding significance to our researches. Revised portion are marked in yellow in the revised manuscript. Please see the attachment
The main corrections in the paper and the responds to the reviewer’s comments are as flowing:
- Comment: The Introduction section should be split into 2 - Introduction and Literature Review; Moreover, the Literature review should aim to present international studies from your field of study as most of your literature is from China;
Response: in line 121-204, we have divided the introductory section into two parts. The literature review focusing on international research in the field of supplementary cropland conservation.
- Comment: In the Game model part, there should be 1-2 paragraphs dedicated to presenting similar papers that apply the same or similar model in the same field or a different one;
Response: In 170-197, we add the detailed application of evolutionary games in the literature review
- Comment: In the Conclusions section I would add some limitations of this study and further research gaps to be tackled.
Response: In 759-767, we revised our conclusions to add the limitations of this study as well as future research directions
- Comment: Fine tuning is needed throughout the entire article - for example, line 453 and line 513 - Figure 3 & 6 (with capital F); line 605 - Suggestions (capital S).
Response: We have revisited the manuscript and revised the errors that occurred.
We appreciate for Reviewers’ warm work earnestly, and hope that the correction will meet with approval.

Reviewer 3 Report
I read the study, and I am convinced of the analysis that came with it, and I recommend its acceptance for publication
In conclusion, this study created an evolutionary game model of local governments, ATSOs, and farmers, concentrated on simulating the evolutionary trend of black soil cultivated land conservation, and examined the influencing factors of stakeholders' behavioural strategies through numerical simulation to present reasonable recommendations for the implementation of cultivated land conservation.
Author Response
Dear Reviewer:
Thank you for your recognition and approval of our manuscript. We have taken the initiative to further optimize the paper. Revised portion are marked in yellow in the revised manuscript. Please see the attachment.
We appreciate for Reviewers’ warm work earnestly, and hope that the correction will meet with approval.

Round 2
Reviewer 1 Report
Authors, incorporated all suggestion and modifications, now revised MS would be accept